# A Personalized Multimodal BCI–Soft Robotics System for Rehabilitating Upper Limb Function in Chronic Stroke Patients

**DOI:** 10.3390/biomimetics10020094

**Published:** 2025-02-07

**Authors:** Brian Premchand, Zhuo Zhang, Kai Keng Ang, Juanhong Yu, Isaac Okumura Tan, Josephine Pei Wen Lam, Anna Xin Yi Choo, Ananda Sidarta, Patrick Wai Hang Kwong, Lau Ha Chloe Chung

**Affiliations:** 1Institute for Infocomm Research, Agency for Science, Technology and Research (A*STAR), 1 Fusionopolis Way, #21-01 Connexis (South Tower), Singapore 138632, Singapore; 2College of Computing and Data Science, Nanyang Technological University, 50 Nanyang Ave., Singapore 39798, Singapore; 3Rehabilitation Research Institute of Singapore, Nanyang Technological University, 50 Nanyang Ave., Singapore 639798, Singapore; 4Department of Rehabilitation Science, Hong Kong Polytechnic University, 11 Yuk Choi Rd, Hung Hom, Hong Kong, China; 5Tan Tock Seng Hospital, 11 Jalan Tan Tock Seng, Singapore 308433, Singapore

**Keywords:** BCI, EEG, NIRS, fNIRS, stroke, rehabilitation, soft robotics

## Abstract

Multimodal brain–computer interfaces (BCIs) that combine electrical features from electroencephalography (EEG) and hemodynamic features from functional near-infrared spectroscopy (fNIRS) have the potential to improve performance. In this paper, we propose a multimodal EEG- and fNIRS-based BCI system with soft robotic (BCI-SR) components for personalized stroke rehabilitation. We propose a novel method of personalizing rehabilitation by aligning each patient’s specific abilities with the treatment options available. We collected 160 single trials of motor imagery using the multimodal BCI from 10 healthy participants. We identified a confounding effect of respiration in the fNIRS signal data collected. Hence, we propose to incorporate a breathing sensor to synchronize motor imagery (MI) cues with the participant’s respiratory cycle. We found that implementing this respiration synchronization (RS) resulted in less dispersed readings of oxyhemoglobin (HbO). We then conducted a clinical trial on the personalized multimodal BCI-SR for stroke rehabilitation. Four chronic stroke patients were recruited to undergo 6 weeks of rehabilitation, three times per week, whereby the primary outcome was measured using upper-extremity Fugl-Meyer Motor Assessment (FMA) and Action Research Arm Test (ARAT) scores on weeks 0, 6, and 12. The results showed a striking coherence in the activation patterns in EEG and fNIRS across all patients. In addition, FMA and ARAT scores were significantly improved on week 12 relative to the pre-trial baseline, with mean gains of 8.75 ± 1.84 and 5.25 ± 2.17, respectively (mean ± SEM). These improvements were all better than the Standard Arm Therapy and BCI-SR group when retrospectively compared to previous clinical trials. These results suggest that personalizing the rehabilitation treatment leads to improved BCI performance compared to standard BCI-SR, and synchronizing motor imagery cues to respiration increased the consistency of HbO levels, leading to better motor imagery performance. These results showed that the proposed multimodal BCI-SR holds promise to better engage stroke patients and promote neuroplasticity for better motor improvements.

## 1. Introduction

Stroke is the second leading cause of death and the third most prevalent cause of disability globally, and it imposes a substantial burden on healthcare systems worldwide [1]. Among the primary disabilities encountered by stroke survivors, motor impairment stands out, affecting approximately one-third of patients, resulting in persistent disabilities [2]. Notwithstanding these challenges, studies delving into the human central nervous system have unveiled inherent plasticity abilities that contribute to recovery through rehabilitation interventions [3]. Leveraging these findings, recent clinical research has investigated the use of non-invasive electroencephalography (EEG)-based brain–computer interfaces (BCIs) in post-stroke rehabilitation [4,5,6,7,8].

While studies have demonstrated the clinical efficacy of BCI interventions in stroke rehabilitation, most trials have implemented a standardized intervention design for all stroke patients. We hypothesized that a precise, personalized approach to stroke rehabilitation that is tailored to individual patients holds greater promise than a generic “one-size-fits-all” strategy [9]. In a recent clinical trial [5], a novel BCI-stroke rehabilitation (BCI-SR) intervention incorporated six different activities of daily living (ADL)-oriented tasks. Stroke patients participating in the trial reported positive impacts from these task-specific training exercises. However, all patients received the same intervention involving all six tasks, regardless of their specific impairments or abilities. This standardized approach neglects the diverse needs and profiles of individual patients. The existing literature on personalized stroke rehabilitation research remains limited. Mawson et al. [10] introduced a Personalized Self-Managed rehabilitation System (PSMrS) for stroke, utilizing an integrated insole sensor that allowed patients to set personalized goals or activities. However, this personalized setting did not fundamentally address the specific disability of each patient.

To bridge this research gap, we propose an effective personalized stroke rehabilitation approach for BCI interventions based on the Rehabilitation Research Institute of Singapore (RRIS) Ability Data [11], a large Asian-centric kinematic and kinetic database of healthy participants aged between 21 and 80 years old performing upper limb and lower limb movement tasks, aiming to optimally address the unique needs of each stroke patient.

In addition to this EEG-based BCI-SR intervention, we explore the potential of multimodal BCIs integrating EEG and near-infrared spectroscopy (NIRS) signals. NIRS is a non-invasive method for real-time neuroimaging, quantifying brain activity by measuring blood oxygenation [12,13]. When associated with functional neuroimaging, this technique is known as functional near-infrared spectroscopy, or fNIRS. Near-infrared light in the 700–1300 nm range can deeply penetrate living tissue, including skin and bone, while still interacting with hemoglobin present in the blood [14,15]. Employing the modified Beer–Lambert law [15], fNIRS systems estimate changes in concentrations of oxygenated hemoglobin (HbO) and deoxygenated hemoglobin (HbR) through the absorption of near-infrared light. While fNIRS measures neural activity indirectly with a delayed and lower temporal resolution compared to EEG [16], it is more robust in the presence of electrical noise sources, such as power lines and muscle activity [17,18]. Thus, fNIRS serves as a complementary tool to EEG, allowing researchers to record neurophysiological phenomena which occur on different time scales [18,19,20].

Relying on spectroscopic blood oxygenation readings presents challenges due to periodic changes in hemoglobin oxygenation (HbO) during the respiratory cycle, which can affect fNIRS readings [21,22,23]. This may lead to inconsistent baselines in many fNIRS applications, potentially causing unreliable outcomes like false positives and false negatives [24]. Additionally, respiration-related oscillations in HbO/HbR share a frequency range with hemodynamic response functions, so it is very challenging to remove them using a band-pass filter alone [25]. To address this, we propose using a breathing sensor to initiate each trial, ensuring synchronization with the participant’s respiratory cycle phase [26,27].

Lastly, we aim to integrate this fNIRS-BCI system with a robotics system for stroke rehabilitation. Therapeutic robotic systems for stroke rehabilitation are an emerging technology that can assist stroke patients with their recovery process while lowering the workloads of caregivers and medical staff [28,29]. They come in a variety of forms and functions, including systems that are aimed at improving the precision of lower limb movement trajectories, systems that accurately track lower limb joint torques, and systems for training push–pull motions with the upper limbs [30,31,32]. For this experiment, we used a soft robotics glove system that can assist with grasping motions [7,33].

This study aims to enhance a BCI-SR system through two key components: the introduction of personalized tasks targeting individual patient disabilities and the incorporation of fNIRS as an additional modality improved by synchronizing respiration trial onset timings.

### Two-Stage Development of a Multimodal BCI-SR System

This paper describes two sets of experiments, associated with two stages of development for our BCI-SR system. The first set involved the development of a respiratory-driven multimodal BCI system and conducting in-house trials with healthy participants to enhance the accuracy and reliability of HbO recordings. For this stage, our hypothesis was that respiration (i.e., the inhalation and exhalation cycles associated with normal breathing) would affect HbO levels over time. We also hypothesized that synchronizing trial onset timings with respiration during an fNIRS BCI task would improve the quality of fNIRS recordings.

In the second stage, clinical trials were conducted at Tan Tock Seng Hospital (Singapore) with chronic stroke patients, utilizing the multimodal BCI system developed earlier. Our hypothesis at this stage is that stroke patients using our respiratory-driven multimodal BCI system would see significant motor recovery after rehabilitation.

The primary objective of this study was to evaluate the effectiveness of a personalized stroke rehabilitation approach driven by Ability Data and to enhance motor outcomes in chronic stroke patients. By tailoring interventions to individual needs and leveraging the advantages of the multimodal BCI system, our goal is to optimize the rehabilitation process and improve functional recovery.

## 2. Materials and Methods

This study was divided into two phases. The first involved the development of a respiration-driven multimodal BCI system, which was tested on healthy participants. The second phase involved testing the system on stroke patients and documenting their recovery of motor functions as rehabilitation was ongoing.

### 2.1. Respiration-Driven Multimodal BCI System

The BCI system consisted of a 24-channel EEG system (Neurostyle EEG model D1, made by Neurostyle Pte. Ltd., Singapore, Singapore) and an NIRS instrument called NIRSport2 (NIRx Medizintechnik GmbH, Berlin, Germany). To enable simultaneous EEG and fNIRS data collection, a customized cap was developed. For fNIRS data, we used 8 infrared sources and 8 detectors, using wavelengths of 760 nm and 850 nm. The optodes, positioned following the International 10–20 system above the motor cortex on both hemispheres, resulted in 10 channels per hemisphere with a fixed 3 cm distance between each source–detector pair. The fNIRS and EEG data were acquired at sampling rates of 8.72 Hz and 250 Hz, respectively. A respiration strain sensor (SA9311M Respiration Sensor, ThoughtTech, Montreal, QC, Canada) on the participant’s abdomen facilitated breath detection. Figure 1 illustrates the system’s setup, with the left plot depicting the sensors used in the experiment and the right plot providing an overview of the experimental flowchart.

Motor imagery tasks, such as “move right” and “move left”, were visually cued to the participants and were presented in a randomized order. Participants were instructed to mentally perform the specific limb movement associated with each cue without any actual physical movement. Each motor imagery task was repeated 20 times for each participant. The fNIRS recording was synchronized with the EEG recording.

### 2.2. Soft Robotic Glove Module

The soft robotic glove module was adapted from a previous study [7]. The mBCI model was integrated with a soft robotic (SR) glove module [33], forming what we term the mBCI-SR system. In a typical mBCI-SR trial, the visual feedback UI prompts the participant for motor intent. The participant then engages in motor imagery while EEG and fNIRS signals are collected. Successful motor imagery triggers a signal to both the robotic glove control module and visual feedback system, activating the glove’s actuators and offering visual acknowledgment of the detected motor intent.

The SR module, consisting of a control box and fabric glove, seamlessly responded to BCI-detected motor imagery, offering mechanical feedback. Crafted from fabric, the glove incorporated four dorsal actuators aligned with the fingers, ensuring proficient grasping and manipulation. By addressing common issues in stroke survivors’ thumbs, a thermoset plastic splint enhanced alignment and mobility, enabling optimal hand postures for diverse activities [33]. Figure 2 illustrates a participant operating the soft robotic glove. The control box, equipped with a microcontroller, orchestrated the movement of an air compressor and valves linked to each actuator. The coordination of straightening and bending across all four actuators allowed for a variety of hand postures, such as grasping, 2-finger pinching, or tripod pinching, each applicable to different activities.

### 2.3. Experimental Protocol and Healthy Participants’ Data Collection

In traditional BCI experimental protocols, the onset of motor imagery (MI) cues is not aligned with respiratory patterns. Since HbO levels change over time during respiratory cycles, we hypothesize that this lack of alignment can result in inconsistent HbO baselines recorded by fNIRS and therefore decrease the accuracy of the fNIRS-based BCI. In our study, we introduced an innovative approach by specifically triggering MI cues during the inhalation phase of the respiration cycle (Figure 3). This was achieved using a respiration sensor that detects abdominal expansion or contraction, transmitting respiratory data to the computer via a USB port.

To identify real-time inhalation and exhalation events, we developed a peak detection algorithm tailored to the respiratory cycle. Following a preparatory period, the algorithm detects inhalation and presents the MI cue on the screen, synchronizing it with the detected inhalation event [27].

This in-house study was approved by the Institutional Review Board at A*STAR (Agency for Science, Technology and Research), Singapore (IRB Reference 2021-166, approval obtained on 24 November 2021). Ten healthy adult participants (four females, six males) gave their informed consent and participated in the experiment. Each participant performed trials in 4 experimental blocks containing 40 trials each. Every block contained equal numbers of left and right MI trials. A total of 160 trials were collected from each participant.

### 2.4. Healthy Participants’ fNIRS Data Analysis

Raw fNIRS signals first underwent band-pass filtering using a third-order Butterworth filter (with cutoff frequencies set at 0.05 and 0.60 Hz) to remove systemic artifacts associated with pulses and very slow oscillations. We applied the modified Beer–Lambert law [15] to convert the raw intensity data into changes in HbO and HbR concentrations. The extracted HbO and HbR time series data serve as critical indicators of local oxygen supply and demand, allowing us to discern patterns of changes in neural activity related to cognitive or motor tasks.

By incorporating respiration-synchronized MI cues in our experiment, we anticipated observing more consistent HbO levels at the onset of the MI period, leading to reduced HbO dispersion across the 160 single trials. We hypothesized that this improvement in HbO consistency would enhance the decoding performance of MI. To objectively measure this effect, we compared the distribution of HbO values for two 2 s periods, $B_{1}$ and $B_{2}$, as illustrated in Figure 2B. $B_{1}$ represents HbO values for the respiration-unsynchronized case, commonly used in conventional BCI protocols where MI cues are not triggered by the respiration cycle. $B_{2}$ represents HbO values for the respiration-synchronized case obtained in our experiment, where MI cues were synchronized with the respiration cycle.

We quantified the difference between these two cases by comparing the HbO variance in $B_{1}$ and $B_{2}$ across the 160 trials, defining it as the dispersion index (DI) in Equation (1), given by(1)$$DI=\frac{\sum_{i=1}^{n_{t}}{\left(x_{i}^{B_{2}}-{\bar{x}}^{B_{2}}\right)}^{2}}{\sum_{i=1}^{n_{t}}{\left(x_{i}^{B_{1}}-{\bar{x}}^{B_{1}}\right)}^{2}}$$
where $n_{t}\;$ is the number of trials, $x_{i}^{B_{1}}$ and $x_{i}^{B_{2}}$ are the HbO values of the $i^{th}$ trial from segments $B_{1}$ and $B_{2},$ respectively, and ${\bar{x}}^{B_{1}}$ and ${\bar{x}}^{B_{2}}$ represent the means of the HbO values from segments $B_{1}$ and $B_{2},$ respectively. Changes in HbO and HbR concentrations were calculated from changes in optical density (ΔOD) using the modified Beer–Lambert law. In our analysis, we focused solely on HbO values due to their superior predictive performance compared to HbR values [20]. Subsequently, the HbO data were analyzed separately for left and right motor imagery (MI). Using HbO values from multiple channels across epochs allows for the creation of a spatial–temporal feature space, capturing changes in brain blood saturation. Building on this feature, we developed a straightforward algorithm for motor imagery (MI) detection. The algorithm was composed of three main components: feature scaling and normalization, vectorization, and classification using a Linear Discriminant Analysis (LDA) model. To assess algorithm performance, we employed 10-fold cross-validation for each participant’s data. We note that we specifically evaluated MI performance under the respiration-synchronized method, as opposed to the more commonly used respiration-unsynchronized method.

### 2.5. Clinical Trial for Chronic Stroke Patients

Following the development of the mBCI-SR system and in-house trials, we initiated a clinical trial aimed at rehabilitating post-stroke patients. We obtained ethical approval from the Domain Specific Review Board (DSRB), National Healthcare Group, Singapore. (Reference no: 2021/00715; clinical trial registration number NCT05642299 in ClinicalTrials.gov). All participants provided informed consent prior to enrolment for this study.

The inclusion criteria for participants included an age range of 50–80 years, no limit on lesion size, and a stroke occurrence at least 6 months prior to the clinical trial. Additionally, participants were required to have Fugl-Meyer Assessment (FMA) scores of 11–45 out of 66 and the ability to follow commands and sit upright for 1.5 h. Cognitive intactness and satisfactory performance during BCI screening were also prerequisites. The exclusion criteria were defined to rule out individuals with recurrent stroke, hemi-spatial neglect, severe spasticity, contracture, deformity, significant vision and hearing impairment, or poor skin conditions.

Our team had a target of 10 participants for this study; however, the recruitment period was conducted during the 2020 COVID-19 pandemic in Singapore, resulting in difficulties with recruitment. In total, 5 stroke patients were recruited for this trial, and out of these 5, 1 did not meet the inclusion criteria and was withdrawn from the study.

This was a single-group experiment conducted to investigate the efficacy of the BCI-based soft robotic glove, incorporating individually tailored activities for stroke rehabilitation. The total intervention period spanned 6 weeks, with 3 sessions per week, totaling 18 sessions. Each therapy session comprised a single supervised run and four personalized activities of daily living (ADL)-oriented task runs, with a 3–5 min break between runs. The supervised run aimed to collect additional EEG data for further training the subject-specific model for subsequent therapy sessions. During the ADL-oriented task runs, participants were instructed to perform motor imagery (MI) for activity-specific tasks, with visual feedback provided through a virtual arm and virtual participants depicting six different tasks. The ADL tasks encompassed scanning goods, moving an object upwards to a cabinet, using two hands to move a towel, pouring water into a cup, performing an eating action, and the fine motor movement of picking up a small block using two fingers (see Table A1 in Appendix A).

During the MI task, participants were instructed to imagine moving their stroke-affected arm and fingers to carry out the task. Participants proceeded with the actual ADL task with assistance from the soft robotic glove if MI was detected by the system. No assistance was provided for the participants with regard to their arm movement (specifically, movement of the elbow or shoulder). The only assistance was provided by the soft robotic glove augmenting their finger movements. This was because the design and intent of the ADL tasks was to emphasize the use of fine motor skills involving fingers. Each ADL-oriented task run comprised 40 trials. The participants were given two attempts to perform the respective MI task; if MI was not detected on the first attempt, the system would activate the soft robotic glove on the second attempt. Each trial lasted ∼16 s and each run lasted ∼11 min, adding up to approximately 1.5 h for each therapy session, including the time taken to set up the EEG system.

The outcome measures assessed in this study were the FMA for an upper extremity and the Action Research Arm Test (ARAT). All the participants’ baseline outcome measures were assessed on the 0th week before and after the intervention. A post-intervention assessment was also carried out on the 6th week and 12th week. All assessments were conducted at the hospital by a physiotherapist. A questionnaire for patients’ subjective rating of their respective intervention was also administered post intervention to gather general feedback from the participants on the clinical trials, their self-perceived performance in ADL, and any experiences they might have encountered during the clinical trials.

### 2.6. Personalized Treatment Using RRIS Ability Data

The six ADL tasks incorporated in the BCI-SR intervention were considered a “one-size-fits-all” training intervention for stroke patients. Given the diverse degrees of upper limb impairment among stroke patients, not all ADL tasks may prove equally beneficial or optimized for individual rehabilitation. Thus, we proposed to use normative data (Ability Data) collected by the RRIS as a reference to customize the rehabilitation program, considering individual stroke patients’ impairment characteristics [11].

Six upper limb tasks from the RRIS’s Ability Data—specifically folding a towel towards the participant, folding a towel sideways, forward reaching, lateral transferring, hand-to-mouth movements, and turning a key—were aligned with the six ADLs of the BCI-SR intervention. This alignment was established based on the similarity among the movement trajectories, as detailed in Table A1 in Appendix A. Patients were assigned these upper limb tasks for repetition, and their performance was recorded using the Miqus M3 motion capture system (Qualisys, AB, Sweden). This recording occurred in the motion capture laboratory at the RRIS, where the movements were captured in three-dimensional space.

To ascertain the appropriate ADL tasks for each stroke patient, their movement performances were compared with those of normative participants, involving several key steps. The initial step included segmentation, where each task in the Ability Data was isolated to precisely align with the six ADL tasks identified in Appendix A. This ensured that the evaluation was tailored to the specific requirements of each task. Subsequently, a confidence interval was established by comparing the mean joint angle for each normative participant with the overall mean joint angle of an age-matched normative participant under the Ability Data. Dynamic Time Warping (DTW) was employed for this comparison, resulting in an individual DTW score for each participant, as outlined in Equation (2):(2)$${DS}_{S_{i}}=DTW\left(\frac{\sum_{r}{JA}_{r,s_{i},t}}{max(r)}\forall t,\frac{\sum_{s_{a}}\sum_{r}{JA}_{r,s_{a},t}}{\sum_{s_{a}}max(r)}\forall t,s_{i}{\neq s}_{a}\right),\forall_{S_{i}}$$
where all $s_{a},s_{i}\in$ all age-matched participants.

A confidence interval was derived by pooling all normative ${DS}_{S_{i}}$, providing a range within which the normative movements typically fall. Each stroke patient’s score was calculated using a similar method, as illustrated in Equation (3).(3)$${DS}_{S_{p}}=DTW\left(\frac{\sum_{r}{JA}_{r,s_{p},t}}{max(r)}\forall t,\frac{\sum_{s_{a}}\sum_{r}{JA}_{r,s_{a},t}}{\sum_{s_{a}}max(r)}\forall t\right),\forall_{S_{p}}$$
where $s_{a}\in$ all age-matched normative participants and $s_{p}\in$ all stroke patients.

If the ${DS}_{S_{p}}$ of the stroke patient for all joint movements for any of the six upper limb tasks was within the 95% confidence interval of the age-matched normative movements, it was assumed that the stroke patient’s upper limb movement was similar to the normal movements of a healthy adult, and hence they did not require rehabilitation. The tasks that met the intervention requirements were further analyzed.

As different joints play different roles in a task (for example, turning a key requires more supination and pronation in the forearm as compared to flexion and extension in the shoulder), various joints were assigned different weights as percentages for each specific task, reflecting their contribution to the overall movement. The weightage was calculated based on the normative maximum and minimum range of motion compared to the physiological maximum and minimum range of motions, as shown in Equation (4):(4)$$W_{joint}=Abs\left(\frac{\max({ROM}_{s_{a}})-\min({ROM}_{s_{a}})}{\max({ROM}_{physiological})-\min({ROM}_{physiological})}\right)\times 100\%$$

A normal distribution of normative movement was obtained by tabulating the mean and standard deviation of joint angles for each individual participant. As such, we calculated the z-score of the stroke patients’ movements using the normal distribution of normative movement. Our goal was to obtain a final score for each task, ${FS}_{task}$, that provided a basis for understanding how the stroke patient performed based on the distribution patterns of normative movements at every joint angle, as shown in Equation (5):(5)$${FS}_{task}=\sum_{joint}W_{joint}\times Z-{score}_{joint}$$

This final score was then used to rank and prioritize four out of the six upper limb tasks not within the 95% confidence interval of normative movement values. The matched ADL from the BCI-SR intervention would then form the rehabilitation regime for the stroke patient.

## 3. Results

### 3.1. Brain Activation Topographic Maps Derived from EEG and fNIRS Recordings

Our clinical trial for stroke rehabilitation enrolled five participants. Among them, four patients successfully passed the screening process and completed the entire clinical trial. The trial protocol included calibration and training sessions. In this report, we focus solely on the analysis of the calibration session trials. The calibration session consisted of 4 runs, each comprising 80 trials, totaling 320 trials. Within each run, 40 trials involved MI, and the remaining 40 trials were dedicated to an idle condition. Each trial lasted approximately 12 s, and each run took approximately 8 min.

To explore the brain activity associated with motor imagery tasks, we created topographic brain maps using EEG and fNIRS data separately. The EEG data analysis employed the filter band common spatial pattern (FBCSP), a technique frequently used to enhance the detection of motor imagery patterns by filtering EEG data into specific frequency bands and extracting common spatial patterns related to motor imagery [34,35,36]. The resulting topographical maps visually represent the spatial distribution of brain regions involved in motor imagery tasks. Simultaneously, we utilized Generalized Linear Models (GLMs) to examine the relationship between fNIRS hemodynamic responses under HbO and MI conditions at the participant level. GLM parameters offer a quantitative understanding of the connection between motor imagery conditions and neural responses, considering spatial dynamics across multiple channels or sensors [37,38].

The brain activation and topographic maps succinctly depicted each participant’s specific neural engagement during motor imagery tasks. It is essential to highlight that the epochs used for the EEG data post MI cue spanned 4 s, while for the fNIRS data, the epochs extended to 9 s. This divergence in epoch length reflects the distinct temporal responses from these modalities.

Figure 4 illustrates the topographic brain maps derived from our data analysis. Upon closer inspection of these maps, a coherence emerges in the activation patterns across EEG and fNIRS for all four participants. The consistent activation pattern observed in both modalities lends credence to the notion that they collectively capture the intricate neural representations associated with motor imagery. The overlapping patterns likely indicate a shared neural substrate engaged during motor tasks, emphasizing the complementary nature of EEG and fNIRS in deciphering brain activity. The two modalities together give us insights into the processes occurring in the brain during MI.

### 3.2. Respiration Synchronization Effect

Our prior investigation reported the efficacy of synchronizing the MI cue in an fNIRS BCI experiment to mitigate confounding respiration effects on fNIRS HbO readings [27]. We conducted a more in-depth analysis using data collected from the clinical trial.

We present multi-trial HbO values along a timeline, focusing on patient 3 for illustration, as depicted in Figure 5. It is crucial to highlight the evident impact of respiration synchronization on the result. The pattern of HbO changes exhibits significantly greater consistency across trials in the synchronized cases (middle, bottom) when compared to the unsynchronized cases (top). This observation underscores the importance of respiration synchronization in enhancing the robustness and consistency of HbO values in a clinical setting, further validating the efficacy of our approach.

### 3.3. Motor Imagery Prediction Using fNIRS Data

The extracted HbO and HbR values from fNIRS signals serve as valuable indicators of local oxygen supply and demand, providing insights into regional brain activity patterns associated with MI tasks. We deployed a spatial–temporal estimator that incorporates both temporal information from HbO time series data and spatial information from different channels. This approach was applied to datasets from both healthy participants and stroke patients. For each participant, we conducted 10-fold cross-validation to derive the area under the receiver operating characteristic curve (AUC-ROC) for the spatial–temporal estimator. Our observations revealed that only half of the healthy participants’ fNIRS signals could be translated into MI, while all the stroke patients who passed the screening session exhibited effective MI-informed fNIRS signals.

The prediction results are illustrated in Figure 6, where Figure 6A,B show the prediction results for healthy participants and stroke patients, respectively. In both cases, gray bars represent the mean accuracy for each participant, with standard deviations depicted as error bars. The dotted line indicates a chance accuracy of 0.5. An asterisk signifies that the MI accuracy surpassed the chance level for that participant (one-tailed *t*-test, *α* = 0.005 after Bonferroni correction). In the case of healthy participants, 8 out of 10 participants performed better than the chance level, whereas for the stroke patients’ data, all 4 participants performed better than the chance level. To quantify the respiration synchronization effect, we calculated the dispersion index for each participant and examined the correlation between the dispersion index and MI decoding accuracy. A meta-analysis combining data from both healthy participants and patients revealed a negative correlation between MI accuracy and the dispersion index (Pearson’s correlation coefficient = −0.545), as depicted in Figure 6C.

Table 1 below shows the sensitivity and specificity of the MI decoding algorithm we used in healthy participants and stroke patients.

### 3.4. Effectiveness of Clinical Rehabilitation

We measured the Fugl-Meyer Assessment (FMA) and Action Research Arm Test (ARAT) scores of four stroke patients as they underwent rehabilitation. These are plotted below in Figure 7.

On week 2, the FMA scores were higher than the pre-trial baseline by 8.75 ± 1.84, and the ARAT scores were higher than the pre-trial baseline by 5.25 ± 2.17 (mean ± SEM).

## 4. Discussion

We introduced a breathing sensor into the BCI system, hypothesizing that synchronizing motor imagery cues to respiration would enhance the consistency of hemoglobin oxygenation (HbO) levels. The dispersion index derived from the HbO data indicated decreased variability across the MI trials in 7 out of 10 participants, supporting our hypothesis.

However, despite observing lower HbO variance in the respiration-synchronized case, an increase in the dispersion index was noted in 2 out of the 10 participants. Possible explanations include inherent irregular breathing in these two participants or potential artifacts in the breathing sensor data due to physical movements. While the general incidence of abnormal breathing patterns is not well documented, it is possible that irregular breathing was caused by conditions such as dyspnea, which affects about 10% of the general adult population worldwide [39]. Further analyses of the breathing sensor data, coupled with instructions given to participants for deep and regular breathing, improvements to the inhalation detection algorithm, and the potential implementation of exclusion criteria for participants with excessively irregular breathing patterns, are warranted. Nonetheless, excluding individuals with abnormal breathing patterns from research studies would also adversely affect our efforts to treat stroke patients, as strokes are known to frequently cause abnormal breathing patterns [40,41,42]. The respiratory effects of strokes may confound fNIRS studies on stroke patients. In our opinion, more research is needed on how strokes interact with HbO levels in the brain and fNIRS recordings.

We observed a correlation between a higher dispersion index and poorer motor imagery (MI) performance, emphasizing the importance of obtaining consistent HbO readings without the confounding effects of respiration in fNIRS-based BCIs. The proposed method of synchronizing motor imagery cue timing with the participant’s respiration cycle holds promise for improving the decoding performance of the fNIRS-BCI system by mitigating the effects of respiration. The preliminary evidence presented in this paper aims to guide fNIRS-BCI researchers in deploying similar experimental protocols to reduce respiration-related confounding effects.

Moreover, the integration of EEG and fNIRS enhances our ability to unveil subtle nuances in neural patterns, providing a more comprehensive and robust understanding of underlying brain dynamics during motor tasks. The topographic brain maps derived from both EEG and fNIRS data revealed overlapping patterns in the activation areas across EEG and fNIRS for all three participants, emphasizing the complementary nature of these modalities in deciphering brain activity.

We observed a substantial improvement in prediction performance during the clinical study as compared to the laboratory trial, and this enhancement can be attributed to several factors. First and foremost, the screening session incorporated in the clinical study successfully excluded individuals who were unable to perform motor imagery (MI), leading to a more focused and adept participant pool. Furthermore, patients involved in the clinical study demonstrated a significantly stronger inclination to utilize MI compared to healthy participants in the laboratory trial. In contrast, the motivation levels of healthy participants during the laboratory trial might have been insufficient to effectively engage in MI for task performance. This discrepancy in motivation levels stands out as a potential contributing factor to the observed variation in prediction performance between the two settings. Overall, the improvements in FMA scores in the patients were on a par with those observed in a previous study [8].

Nonetheless, we note that the variance in FMA and ARAT scores was rather high for the stroke patients, as seen in the large error bars in Figure 7. High variance from patient to patient is to be expected in stroke studies, as strokes are highly heterogenous in both the nature of the brain injury and the recovery process [43]. We were unfortunately restricted to only four participants due to the exigencies imposed by the 2020 COVID-19 pandemic, and we recommend that future studies recruit a larger and more diverse group of participants, which would counteract the inherently high variance in stroke cases and lead to more robust statistical inferences.

While robotic stroke rehabilitation systems have been used in many previous studies [7,30,31,32], we need to explore the scalability and cost-effectiveness of this new system if it is to be used in a clinical setting. While fNIRS systems are far cheaper and more mobile than fMRI systems, they are still more expensive and difficult to set up than dry EEG systems. Also, the system still requires a therapist to determine the appropriate exercise(s) for each stroke patient to use. Ultimately, for a new machine learning system to be successful, it is important to obtain buy-in from all involved stakeholders, including patients, their caregivers, clinicians, and public health funding bodies [44], so any follow-up study must include surveys and interviews for these parties to provide feedback on this system and to gauge their willingness to use it.

## 5. Conclusions

We developed a multimodal BCI system capable of synchronizing MI cues with respiration via a breathing sensor. We observed decreased variability in HbO across MI trials in the majority of participants when respiration cycles and trial timings were synchronized. In addition, we observed a correlation between higher HbO variability and poorer MI performance, suggesting that this variability can explain poor BCI performance in some individuals and that synchronizing trial timings with respiration cycles indeed improves the quality of fNIRS signals in the context of BCIs.

Additionally, the integration of EEG and fNIRS signals provides a more comprehensive understanding of neural activity patterns during motor tasks, as illustrated by the overlapping topographic brain maps. This emphasizes the complementary nature of these modalities when decoding brain activity.

The findings of our clinical study complement the findings of our laboratory trial. All our stroke patients exhibited better-than-chance levels of MI decoding when performing the multimodal BCI-SR task, and all exhibited significant improvements in motor function, as quantified by two metrics. Nonetheless, we recognize that a larger sample size is needed in future studies to draw more robust conclusions and to determine if the multimodal BCI-SR system performs better than a non-BCI-SR system.

On the technical side, our ongoing work involves further refinement of the breathing sensor, algorithm improvements, and better participant selection criteria for future studies.

## Figures and Tables

**Figure 1 biomimetics-10-00094-f001:**
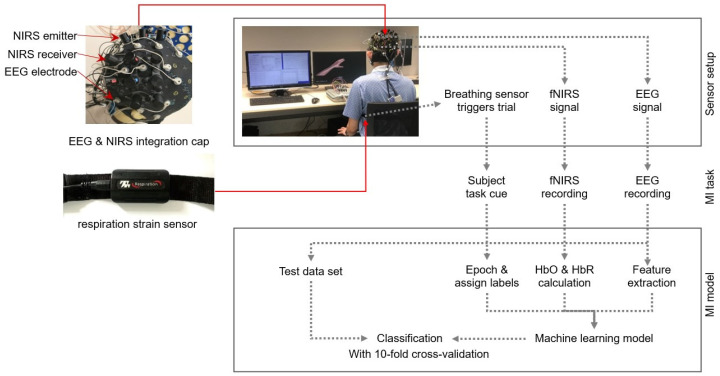
Overview of the experimental setup.

**Figure 2 biomimetics-10-00094-f002:**
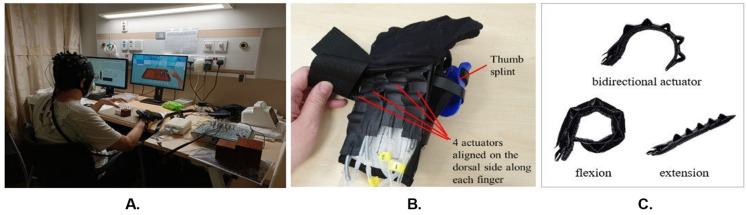
Soft robotic glove module of mBCI-SR system. (**A**) Participant operating the soft robotic glove via motor imagery; (**B**) soft robotic glove and thermoset thumb splint; (**C**) two different modes (flexion, extension) of the bidirectional actuator.

**Figure 3 biomimetics-10-00094-f003:**
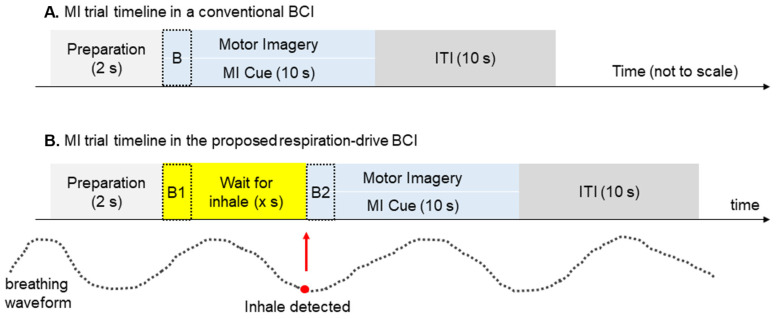
Timelines of motor imagery (MI) trials, not to scale. ITI—inter-trial interval. B, B1, and B2 are baseline segments used for data analysis, explained in the next section. (**A**) In conventional MI experiments, trial timings are not synchronized with the participant’s respiration. After a 2 s preparation period (light gray), a cue is delivered for 10 s (blue), during which participants are instructed to perform MI. Subsequently, there is a 10 s inter-trial interval (ITI) before the next trial (dark gray). (**B**) In our proposed BCI system, the cue to begin MI is only delivered when inhalation is detected after the preparation period. In this example, there is a wait of *x* seconds (yellow).

**Figure 4 biomimetics-10-00094-f004:**
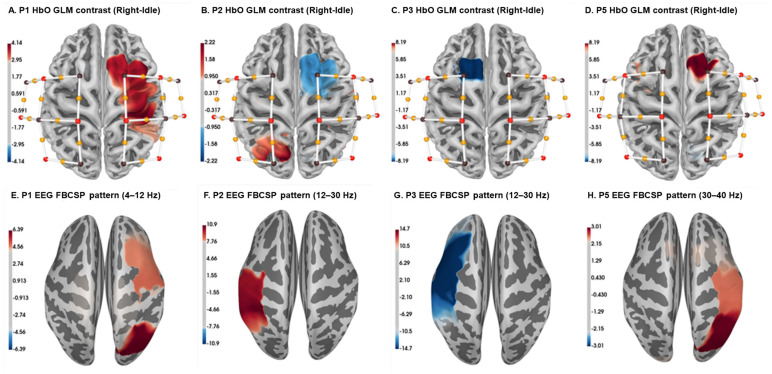
Comparison of topographic brain maps derived from fNIRS GLM analysis and EEG FBCSP approach for the 4 patients’ data. (**A**–**D**): Brain maps derived from HbO data. (**E**–**H**): Brain maps derived from EEG FBCSP data. Each column represents data from the same patient.

**Figure 5 biomimetics-10-00094-f005:**
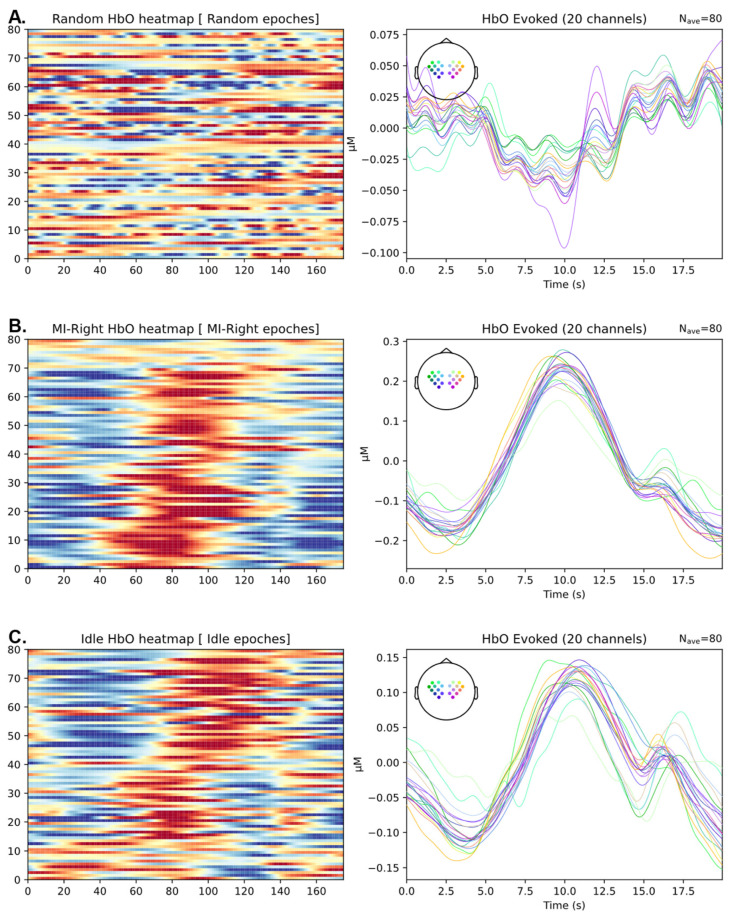
Plot of HbO values for stroke patient No. 3 in the clinical trial. In the left column subplots, each horizontal line in the heatmap represents one trial, with HbO levels quantified by color (measured in units of 10^−7^ mol/L). The subplots in the right column depict the evoked plot, illustrating the average of all 80 trials to present a group effect. (**A**) Eighty randomly selected epochs, indicating how HbO signals look like when not synchronized with any task-related or physiological signals. (**B**) Eighty epochs of HbO recordings, synchronized to the start of right-side motor imagery trials. (**C**) Eighty epochs of HbO recordings, synchronized to the inhalation phase of breathing during idle periods.

**Figure 6 biomimetics-10-00094-f006:**
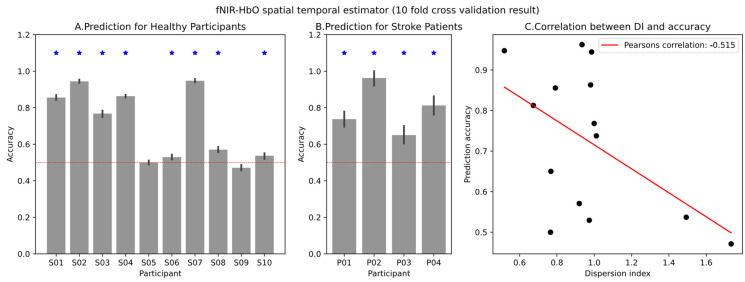
The prediction accuracy of the fNIRS-HbO spatial–temporal estimator and the correlation between the dispersion index and prediction performance. (**A**) Prediction result for healthy group. (**B**) Prediction result for stroke patient group. The blue star on the bar plot shows that the prediction accuracy was significantly better than the chance level (one-tailed *t*-test, *α* = 0.005 after Bonferroni correction). (**C**) Scatter plot showing a negative correlation between prediction accuracy and dispersion index.

**Figure 7 biomimetics-10-00094-f007:**
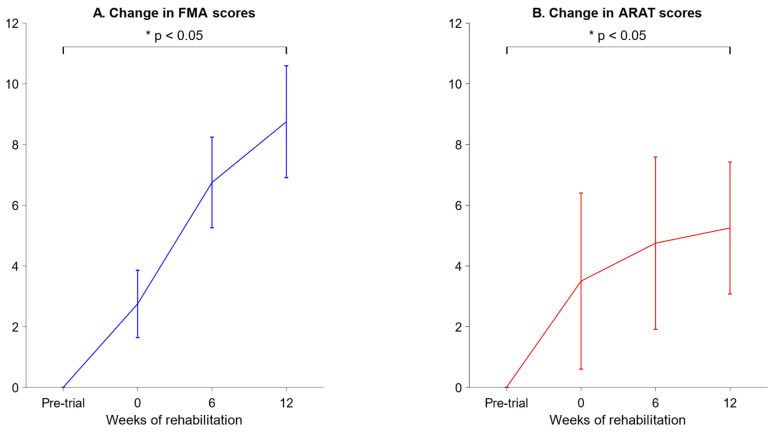
FMA and ARAT scores of stroke patients before and after rehabilitation. (**A**) Changes in FMA scores with respect to the pre-trial baseline. The graph plots the mean change across all 4 patients, with error bars representing the standard error of the mean. The FMA scores on week 12 were significantly higher than the pre-trial scores (indicated by an asterisk, *p* < 0.05, one-tailed paired *t*-test). (**B**) Changes in ARAT scores with respect to the pre-trial baseline. The graph plots the mean change across all 4 patients, with error bars representing the standard error of the mean. The ARAT scores on week 12 were significantly higher than the pre-trial scores (indicated by an asterisk, *p* < 0.05, one-tailed paired *t*-test).

**Table 1 biomimetics-10-00094-t001:** Sensitivity, specificity, and accuracy of the motor imagery decoding algorithm among participants.

Participant Group	Sensitivity	Specificity	Accuracy
Healthy	0.683	0.701	0.692
Stroke patients	0.769	0.809	0.790

## Data Availability

The datasets presented in this article are not readily available because of patient privacy interests. Requests to access the datasets should be directed to the corresponding author.

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
