# Peer review of "A Personalized Multimodal BCI–Soft Robotics System for Rehabilitating Upper Limb Function in Chronic Stroke Patients"

_biomimetics, 2025, doi:10.3390/biomimetics10020094_

Round 1
Reviewer 1 Report
Comments and Suggestions for Authors
The authors have proposed a BCI system. They showed using a breathing sensor to synchronize motor imagery cues with the participant’s respiratory cycle. The work is novel and the manuscript is solid. I believe the results are considered significant for researchers in the field. The topic is in the scope of this journal. I only have a few minor comments to help increasing the quality of the manuscript.
- Panels in Figures 4 and 5 need labels (a, b, c, …). The figure captions need to be revised accordingly as well.
- Labels need to be consistent. Please revise labels in Figures 6 and 7 to be lowercase, and so consistent with labels in other figures.
- The Conclusions section needs to be revised. It can be written much better.
- The error bars in Figure 7 are too large. How authors explain it?
Author Response
We appreciate the reviewers for your precious time reviewing our paper and providing valuable comments. It was your valuable and insightful comments that led to possible improvements in the current version. The authors have carefully considered the comments and addressed each of them. We hope the manuscript meets your high standards after careful revisions. The authors welcome further constructive comments if any.
Below we provide the point-by-point responses. Comments are indicated in bold while replies are in bold purple.
- The authors have proposed a BCI system. They showed using a breathing sensor to synchronize motor imagery cues with the participant’s respiratory cycle. The work is novel and the manuscript is solid. I believe the results are considered significant for researchers in the field. The topic is in the scope of this journal. I only have a few minor comments to help increasing the quality of the manuscript.
Thank you very much for the encouraging comments.
- Panels in Figures 4 and 5 need labels (a, b, c, …). The figure captions need to be revised accordingly as well.
Thank you for the comment, we have labelled Figures 4 and 5 with A, B, C, … and revised the captions to include the descriptions.
(Changes to manuscript below)
Figure 4. Comparison of topographic brain maps derived from fNIRS GLM analysis and EEG FBCSP approach for the 4 patients’ data. (A-D): Brain maps derived from HbO data. (E-H): Brain maps derived from EEG FBCSP data. Each column represents data from the same patient.
Figure 5. HbO values plot for stroke patient No. 3 in the clinical trial. In the left column subplots, each horizontal line in the heatmap represents one trial, with HbO levels quantified by color (measured in units of 10-7 mol/L). The subplots in the right column depict the evoked plot, illustrating the average of all 80 trials to present a group effect. (A) 80 randomly selected epochs, indicating how HbO signals look like when not synchronized with any task-related or physiological signals. (B) 80 epochs of HbO recordings, synchronized to the start of right-side motor imagery trials. (C) 80 epochs of HbO recordings, synchronized to the inhalation phase of breathing during idle periods.
- Labels need to be consistent. Please revise labels in Figures 6 and 7 to be lowercase, and so consistent with labels in other figures.
Thank you for the comment, we have edited all the figure labels and figure captions to consistently use the same case.
- The Conclusions section needs to be revised. It can be written much better.
Thank you for the feedback, we have revised conclusion.
(Changes to manuscript below)
- Conclusions
We developed a multi-modal BCI system capable of synchronizing MI cues with respiration via a breathing sensor. We observed decreased variability in HbO across MI trials in the majority of participants when respiration cycles and trial timings were synchronized. In addition, we observed a correlation between a higher HbO variability and poorer MI performance, suggesting that this variability can explain poor BCI performance in some individuals, and that synchronizing trial timings with respiration cycles indeed improves the quality of fNIRS signals in the context of BCI.
Additionally, the integration of EEG and fNIRS signals provides a more comprehensive understanding of neural activity patterns during motor tasks, as illustrated by the overlapping topographic brain maps. This emphasizes the complementary nature of these modalities when decoding brain activity.
The findings of our clinical study complement the findings of our laboratory trial. All our stroke patients exhibited better-than-chance levels of MI decoding when performing the multi-modal BCI-SR task, and all exhibited significant improvements in motor function, as quantified by two metrics. Nonetheless, we recognize that a larger sample size is needed in future studies to draw more robust conclusions, and to determine if the multimodal BCI-SR system performs better than a non-BCI SR system.
On the technical side, our ongoing work involves further refinement of the breathing sensor, algorithm improvements, and participant selection criteria for future studies.
- The error bars in Figure 7 are too large. How authors explain it?
Thank you for the comment, we have added the explanation on the error bars in the discussion section.
(Changes to manuscript below)
- Discussion
Nonetheless, we do note that the variance in FMA and ARAT scores was rather high for the stroke patients, as seen in the large error bars of Figure 7. High variance from patient to patient is to be expected in stroke studies, as strokes are highly heterogenous in both the nature of the brain injury and the recovery process [43]. We were unfortunately restricted to only 4 participants, due to the exigencies imposed by the 2020 COVID pandemic, and we recommend that future studies recruit a larger and more diverse group of participants, which would counteract the inherent high variance of stroke cases, and lead to more robust statistical inferences.
Reviewer 2 Report
Comments and Suggestions for Authors
The manuscript presents a personalized multimodal Brain-Computer Interface combined with a soft robotics system aimed at rehabilitating upper limb function in chronic stroke patients. The proposed system incorporates a novel approach by tailoring rehabilitation to each patient's specific needs using data from the RRIS Ability Database. This approach contrasts with conventional one-size-fits-all methods and seeks to address individual disabilities more effectively. A key innovation is the synchronization of motor imagery cues with the respiratory cycle, which is hypothesized to enhance the consistency and reliability of Near-Infrared Spectroscopy signals. The integration of EEG and NIRS provides complementary insights into neural activity, offering a more robust understanding of brain dynamics during motor tasks. Clinical trials conducted with chronic stroke patients demonstrated significant improvements in motor function, as evidenced by increased Fugl-Meyer Assessment and Action Research Arm Test scores. These results underscore the potential of the multimodal system to promote neuroplasticity and optimize rehabilitation outcomes.
While the study has notable strengths, there are areas for improvement. The relatively small sample size of four patients in the clinical trial limits the generalizability of the findings. Recruitment challenges during the COVID-19 pandemic are acknowledged, but future studies should aim for larger and more diverse participant groups. There is also limited discussion on the scalability and cost-effectiveness of implementing such a personalized system in routine clinical settings. Methodologically, the manuscript does not address potential biases introduced by excluding participants with irregular breathing patterns, which could limit the applicability of the respiration-synchronized protocol to a broader population. Additionally, more detailed analysis of the algorithm's performance, including sensitivity and specificity metrics, would strengthen the claims of improved motor imagery prediction. Despite these limitations, the work makes a significant contribution to the field of stroke rehabilitation and offers a promising framework for integrating personalized and multimodal approaches into neurorehabilitation practices.
Author Response
We appreciate the reviewers for your precious time reviewing our paper and providing valuable comments. It was your valuable and insightful comments that led to possible improvements in the current version. The authors have carefully considered the comments and addressed each of them. We hope the manuscript meets your high standards after careful revisions. The authors welcome further constructive comments if any.
Below we provide the point-by-point responses. Comments are indicated in bold while replies are in bold purple.
- The manuscript presents a personalized multimodal Brain-Computer Interface combined with a soft robotics system aimed at rehabilitating upper limb function in chronic stroke patients. The proposed system incorporates a novel approach by tailoring rehabilitation to each patient's specific needs using data from the RRIS Ability Database. This approach contrasts with conventional one-size-fits-all methods and seeks to address individual disabilities more effectively. A key innovation is the synchronization of motor imagery cues with the respiratory cycle, which is hypothesized to enhance the consistency and reliability of Near-Infrared Spectroscopy signals. The integration of EEG and NIRS provides complementary insights into neural activity, offering a more robust understanding of brain dynamics during motor tasks. Clinical trials conducted with chronic stroke patients demonstrated significant improvements in motor function, as evidenced by increased Fugl-Meyer Assessment and Action Research Arm Test scores. These results underscore the potential of the multimodal system to promote neuroplasticity and optimize rehabilitation outcomes.
Thank you very much for the encouraging comments.
- While the study has notable strengths, there are areas for improvement. The relatively small sample size of four patients in the clinical trial limits the generalizability of the findings. Recruitment challenges during the COVID-19 pandemic are acknowledged, but future studies should aim for larger and more diverse participant groups.
Thank you for the comment, we have added the limitation on the small sample size and the generalizability to the discussion.
(Changes to manuscript below)
- Discussion
Nonetheless, we do note that the variance in FMA and ARAT scores was rather high for the stroke patients, as seen in the large error bars of Figure 7. High variance from patient to patient is to be expected in stroke studies, as strokes are highly heterogenous in both the nature of the brain injury and the recovery process [43]. We were unfortunately restricted to only 4 participants, due to the exigencies imposed by the 2020 COVID pandemic, and we recommend that future studies recruit a larger and more diverse group of participants, which would counteract the inherent high variance of stroke cases, and lead to more robust statistical inferences.
- There is also limited discussion on the scalability and cost-effectiveness of implementing such a personalized system in routine clinical settings.
Thank you for the comment, we added details on the feasibility of implementing this stroke rehabilitation system.
(Changes to manuscript below)
- Discussion
While robotic stroke rehabilitation systems have been used in many previous studies [7,30–32], we need to explore the scalability and cost-effectiveness of this new system if it is to be used in a clinical setting. While fNIRS systems are far cheaper and more mobile than fMRI systems, they are still more expensive and difficult to set up than dry EEG systems. Also, the system still requires a therapist to determine the appropriate exercise(s) for each stroke patient to use. Ultimately, for a new machine learning system to be successful, it is important to obtain buy-in from all involved stakeholders, including patients, their caregivers, clinicians, and public health funding bodies [44], so any follow-up study must include surveys and interviews for these parties to provide feedback on this system, and to gauge their willingness to use it.
- Methodologically, the manuscript does not address potential biases introduced by excluding participants with irregular breathing patterns, which could limit the applicability of the respiration-synchronized protocol to a broader population.
Thank you for the comment, we have added text to discuss how abnormalities in breathing may confound fNIRS studies.
(Changes to manuscript below)
- Discussion
We introduced a breathing sensor into the BCI system, hypothesizing that synchronizing motor imagery cues to respiration would enhance the consistency of Hemoglobin Oxygenation (HbO) levels. The dispersion index derived from HbO data indicated decreased variability across MI trials in 7 out of 10 participants, supporting our hypothesis.
However, despite observing lower HbO variance in the respiration-synchronized case, an increase in the dispersion index was noted in 2 of the 10 participants. Possible explanations include inherent irregular breathing in these two participants or potential artifacts in the breathing sensor data due to physical movements. While the general incidence of abnormal breathing patterns is not well documented, it is possible that irregular breathing was caused by conditions such as dyspnea, which affects about 10% of general adult populations worldwide [39]. Further analysis of the breathing sensor data, coupled with instructions to participants for deep and regular breathing, improvements to the inhalation detection algorithm, and potential implementation of exclusion criteria for participants with excessively irregular breathing patterns, are warranted. Nonetheless, excluding individuals with abnormal breathing patterns from research studies would also adversely affect our efforts to treat stroke patients, as strokes are known to frequently cause abnormal breathing patterns [40–42]. The respiratory effects of strokes may confound fNIRS studies in stroke patients. In our opinion, more research is needed on how strokes interact with HbO levels in the brain and fNIRS recordings.
- Additionally, more detailed analysis of the algorithm's performance, including sensitivity and specificity metrics, would strengthen the claims of improved motor imagery prediction.
Thank you for the comment, we have added details on the sensitivity and specificity of the motor imagery classification algorithm.
(Changes to manuscript below)
Table 1 below shows the sensitivity and specificity of the MI decoding algorithm we used, in healthy participants and stroke patients.
Table 1. Sensitivity, specificity, and accuracy of the motor imagery decoding algorithm among participants.
|
Participant group |
Sensitivity |
Specificity |
Accuracy |
|
Healthy |
0.683 |
0.701 |
0.692 |
|
Stroke patients |
0.769 |
0.809 |
0.790 |
- Despite these limitations, the work makes a significant contribution to the field of stroke rehabilitation and offers a promising framework for integrating personalized and multimodal approaches into neurorehabilitation practices.
Thank you very much for the encouraging comment.
Reviewer 3 Report
Comments and Suggestions for Authors
The article proposes a NIRS-based brain-computer interface system with a soft robotics (BCI-SR) component for personalized stroke rehabilitation. The clinical trial on the personalized multimodal BCI-SR for stroke rehabilitation showed that the proposed multimodal BCI-SR holds promise to better engage stroke patients and promoted neuroplasticity towards better motor improvements. The following issues need to be addressed carefully.
1. The structural level of the first part of the article is confusing, and the analysis of the current research status is not enough. It is recommended that the authors add the relevant content, and it is also recommended that the authors put the research methodology of the multi-modal BCI-SR system proposed in the article in the second part of the article, so as to enhance the rationality of the article.
2. The authors propose an innovative method in the experimental protocol in section 2.2 of the article, which detects abdominal dilation by using a respiratory sensor and transmits respiratory data to a computer via a USB port. However it lacks a detailed description of the method, such as the reasons for proposing the method and the advantages of the method, etc. It is suggested that the authors add relevant content.
3. The structure of chapter 2 of the article is rather disorganized. For example, whether parts 2.5 and 2.6 are experimental protocols and what is the significance of placing them in chapter 2? The authors have described the experimental protocols in sections 2.1 and 2.2. The contents of sections 2.5 and 2.6 do not match the title of chapter 2, it is recommended that the authors should revise the structure and hierarchy of the second part.
4. In "introduction" section, I feel the current coverage of the state of the art is not satisfactory as the related work section does not cover many contributions that likely provide the building blocks of the proposed approach. For example, a. Adaptive human-robot interaction torque estimation with high accuracy and strong tracking ability for a lower limb rehabilitation robot. b. Adaptive patient-cooperative compliant control of lower limb rehabilitation robot. c. A new active rehabilitation training mode for upper limbs based on Tai Chi Pushing Hands.
5. Chapter 4 of the article proposes the introduction of breathing sensors into a biometric (BCI) system, assuming that synchronizing motion picture cues with breathing improves the consistency of hemoglobin, but the dispersion index increased in 2 out of 10 participants, for which the authors proposed an analysis but did not experimentally validate it. It is recommended that the authors design experiments to perform the relevant validation and to enhance the rigor of the article.
6. Picture 4 of the article is not very clear and the author is advised to revise it.
Comments on the Quality of English Language
The English could be improved to more clearly express the research.
Round 2
Reviewer 3 Report
Comments and Suggestions for Authors
The authors have answers the comments well, and the review has no further comment. Thank you.